# Population rate-coding predicts correctly that human sound localization depends on sound intensity

Antje Ihlefeld[1]*, Nima Alamatsaz[1,2], Robert M Shapley[3]

[1]New Jersey Institute of Technology, Newark, United States; [2]Rutgers University, Newark, United States; [3]New York University, New York, United States

**Abstract** Human sound localization is an important computation performed by the brain. Models of sound localization commonly assume that sound lateralization from interaural time differences is level invariant. Here we observe that two prevalent theories of sound localization make opposing predictions. The labelled-line model encodes location through tuned representations of spatial location and predicts that perceived direction is level invariant. In contrast, the hemispheric-difference model encodes location through spike-rate and predicts that perceived direction becomes medially biased at low sound levels. Here, behavioral experiments find that softer sounds are perceived closer to midline than louder sounds, favoring rate-coding models of human sound localization. Analogously, visual depth perception, which is based on interocular disparity, depends on the contrast of the target. The similar results in hearing and vision suggest that the brain may use a canonical computation of location: encoding perceived location through population spike rate relative to baseline.

DOI: https://doi.org/10.7554/eLife.47027.001

**\*For correspondence:**
antje.ihlefeld@njit.edu

**Competing interests:** The authors declare that no competing interests exist.

## Introduction

A fundamental question of human perception is how we perceive target locations in space. Through our eyes and skin, the activation patterns of sensory organs provide rich spatial cues. However, for other sensory dimensions, including sound localization and visual depth perception, spatial locations must be computed by the brain. For instance, interaural time differences (ITDs) of the sounds reaching the ears allow listeners to localize sound in the horizontal plane. In the ascending mammalian auditory pathway, the first neural processing stage where ITDs are encoded, on the timescale of microseconds, is the medial superior olive (MSO). Here, temporally precise binaural inputs converge, and their ITDs are converted to neural firing rate (*Goldberg and Brown, 1968*; *Yin and Chan, 1990*; *Spitzer and Semple, 1995*; *Pecka et al., 2010*; *Day and Semple, 2011*). The shape of the MSO output firing rate curves as a function of ITD resembles that of a cross-correlation operation on the inputs to each ear (*Batra and Yin, 2004*). How this information is interpreted downstream of the MSO has led to the development of conflicting theories on the neural mechanisms of sound localization in humans. One prominent neural model for sound localization, originally proposed by *Jeffress (1948)*, consists of a labelled line of coincidence detector neurons that are sensitive to the binaural synchronicity of neural inputs from each ear, with each neuron maximally sensitive to a specific magnitude of ITD (*Figure 1A*). This labelled-line model is computationally equivalent to a neural place-code based on bandlimited cross-correlations of the sounds reaching both ears (*Domnitz and Colburn, 1977*). Several studies support the existence of labelled-line neural place-code mechanisms in the avian brain (*Carr and Konishi, 1988*; *Overholt et al., 1992*), and versions of it have successfully been applied in many engineering applications predicting human localization performance (e.g.

**eLife digest** Being able to localize sounds helps us make sense of the world around us. The brain works out sound direction by comparing the times of when sound reaches the left versus the right ear. This cue is known as interaural time difference, or ITD for short. But how exactly the brain decodes this information is still unknown.

The brain contains nerve cells that each show maximum activity in response to one particular ITD. One idea is that these nerve cells are arranged in the brain like a map from left to right, and that the brain then uses this map to estimate sound direction. This is known as the Jeffress model, after the scientist who first proposed it. There is some evidence that birds and alligators actually use a system like this to localize sounds, but no such map of nerve cells has yet been identified in mammals. An alternative possibility is that the brain compares activity across groups of ITD-sensitive nerve cells. One of the oldest and simplest ways to measure this is to compare nerve activity in the left and right hemispheres of the brain. This readout is known as the hemispheric difference model.

By analyzing data from published studies, Ihlefeld, Alamatsaz, and Shapley discovered that these two models make opposing predictions about the effects of volume. The Jeffress model predicts that the volume of a sound will not affect a person's ability to localize it. By contrast, the hemispheric difference model predicts that very soft sounds will lead to systematic errors, so that for the same ITD, softer sounds are perceived closer towards the front than louder sounds. To investigate this further, Ihlefeld, Alamatsaz, and Shapley asked healthy volunteers to localize sounds of different volumes. The volunteers tended to mis-localize quieter sounds, believing them to be closer to the body's midline than they actually were, which is inconsistent with the predictions of the Jeffress model.

These new findings also reveal key parallels to processing in the visual system. Visual areas of the brain estimate how far away an object is by comparing the input that reaches the two eyes. But these estimates are also systematically less accurate for low-contrast stimuli than for high-contrast ones, just as sound localization is less accurate for softer sounds than for louder ones. The idea that the brain uses the same basic strategy to localize both sights and sounds generates a number of predictions for future studies to test.

DOI: https://doi.org/10.7554/eLife.47027.002

*Durlach, 1963*; *Hafter, 1971*; *Stern and Trahiotis, 1995*; *Breebaart et al., 2001*; *Hartmann et al., 2005*).

A growing literature proposes an alternative to the labelled-line model to explain mammalian sensitivity to ITD (*Lee and Groh, 2014*). One reason for an alternative is that two excitatory inputs should suffice to implement the labelled-line model, but evidence from experiments on Mongolian gerbils shows that in addition to bilateral excitatory inputs, sharply tuned bilateral inhibitory inputs to the MSO play a crucial role in processing ITDs (*Brand et al., 2002*). Moreover, to date no labelled-line type neurons encoding auditory space have been discovered in a mammalian species. Indeed, using a population rate-code, several studies proposed that mammalian sound localization can be modeled based on differences in firing rates across the two populations of neurons that are tuned to opposing hemispheres (*Figure 1B*; *van Bergeijk, 1962*; *McAlpine and Grothe, 2003*; *Devore et al., 2009*). Rate-based models generally predict that neuronal responses carry most information at the steepest slopes of neural-discharge-rate versus ITD curves, where neural discharge changes most strongly (*Stecker et al., 2005*), consistent with the observation that the peak ITDs of rate-ITD curves often fall outside the physiologically plausible range (*McAlpine and Grothe, 2003*; *Grothe et al., 2010*; but see also *Joris et al., 2006*). In addition, some authors have suggested that how mammalian sound localization adapts to stimulus history further supports a rate-based neural population code, as assessed behaviorally or via magnetoencephalography (*Phillips and Hall, 2005*; *Stange et al., 2013*; *Salminen et al., 2010*).

It is unknown which of the two competing models, broadly characterized as labelled-line versus rate-code model, describes human sound localization better. Here, we observe that the two different models predict different dependencies of sound localization on sound intensity. By combining behavioral data on sound intensity dependence in normal-hearing listeners with numerical

predictions of human sound lateralization from both models, we attempt to disentangle whether human auditory perception is based on a place-code, akin to the labelled-line model, or whether it is instead more closely described by a population rate-code.

An extensive physiology literature characterizes labelled-line versus population-rate type neurons and suggests that, at least from the perspective of evolution, birds and mammals use different neural mechanisms to calculate sound direction (review: *Grothe, 2003*). Thus, we searched the avian and mammalian physiology literature and identified two studies that characterized labelled-line versus population rate-code neurons at low sound levels and as a function of both sound level and ITD (*Peña et al., 1996*; *Zwiers et al., 2004*). Both *Peña et al. (1996)* and *Zwiers et al. (2004)* report neural firing rate in response to acoustic noise stimuli and are thus suitable for predicting each model's sensitivity to the acoustic noises we tested in the current study. Here, we ran a meta-analysis, reconstructing simulated neurons with response characteristics from each of the two studies and using maximum likelihood estimation to predict source laterality from these previous findings.

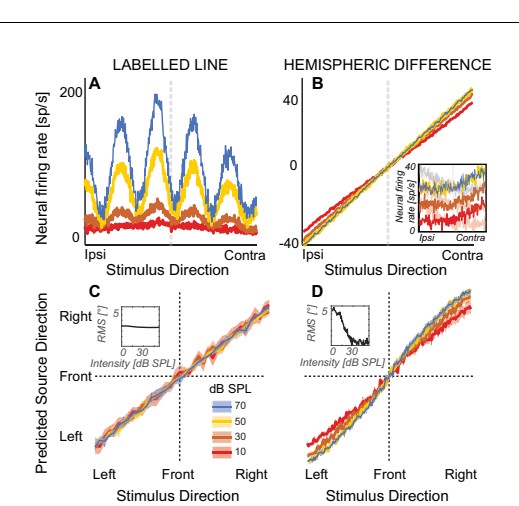

**Figure 1.** Modeling results. (**A**) Firing rate of a simulated *nucleus laminaris* neuron with a preferred ITD of 375 μs, as a function of source ITD. The model predicts source laterality based on the locus of the peak of the firing rate function. (**B**) Hemispheric differences in firing rates, averaged across all 81 simulated *inferior colliculus* units. Rate models assume that source laterality is proportional to firing rate, causing ambiguities at the lowest sound intensities. Inset: Reconstructed responses of an *inferior colliculus* unit. The unit predominantly responds contralaterally to the direction of sound (high-contrast traces). The hemispheric difference model subtracts this activity from the average rate on the ipsilateral side (example shown with low-contrast traces). (**C**) Mean population response using labelled-line coding across a range of ITDs and sound intensities. Inset: The root-mean square (RMS) difference relative to estimated angle at 80 dB SPL does not change with sound intensity, predicting that sound laterality is intensity invariant. (**D**) Mean population response using hemispheric-difference coding. For lower sound intensities, predicted source direction is biased towards midline (compare red and orange versus blue or yellow). For higher sound intensities, predicted source direction is intensity invariant (blue on top of yellow line). Inset: RMS difference relative to estimated angle at 80 dB SPL decreases with increasing sound intensity, predicting that sound laterality is not intensity invariant. Ribbons show one standard error of the mean across 100 simulated responses. Sound intensity is denoted by color (see color key in the figure).

DOI: https://doi.org/10.7554/eLife.47027.003

## Results

### Model Predictions

To predict how lateralization depends on sound intensity from the responses of labelled-line neurons, we estimated neural firing rates from previous recordings in the nucleus laminaris in barn owl (*Peña et al., 1996*). To estimate lateralization's dependence on level based on a population rate-code, we used previous recordings from the *inferior colliculus* of rhesus macaque monkey and calculated hemispheric differences in firing rate (*Zwiers et al., 2004*). The labelled-line neurons predicted that, as sound intensity decreases, perceived source laterality would converge towards similar means for low versus high sound intensities, with increased response variability at decreasing sound intensities (*Figure 1C*). In contrast, the hemispheric-difference model predicted that as sound intensity decreases to near threshold levels, perceived laterality would become increasingly biased toward the midline reference (*Figure 1D*, for example note the shallower slope and thus compressed laterality percepts for red versus blue curves). At higher overall sound intensities, both models predicted that lateralization would be intensity invariant (see insets in *Figure 1C* versus D). Therefore, analyzing how sound intensity affects perceived sound direction near sensation threshold offers an opportunity to disentangle whether our human auditory system relies on a place-based or rate-based population code for localizing sound based on ITD.

A listener's ability to discriminate ITD can vary with sound intensity (*Dietz et al., 2013*). However, it is difficult to interpret previous findings linking sensitivity to ITD and a listener's judgement of sound source direction as a function of sound intensity. Some reported decreased perceived source laterality near sensation threshold (*Teas, 1962*; *Sabin et al., 2005*), but others reported weak or no level effects on perceived lateralization (*Von Békésy and Wever, 1960*; *Mickunas, 1963*; *Hartmann and Rakerd, 1993*; *Macpherson and Middlebrooks, 2000*; *Inoue, 2001*; *Vliegen and Van Opstal, 2004*; *Brungart and Simpson, 2008*; *Gai et al., 2013*). Several factors complicate the interpretation of these previous findings in the context of the current hypothesis. For instance, assuming an approximately 30 dB dynamic range of rate-level function either at the MSO or downstream in the binaural pathway (e.g. *medial superior olive*: *Goldberg and Brown, 1968*; *inferior colliculus*: *Zwiers et al., 2004*), for stimuli at higher sensation levels (SL) where the rate-level functions saturate, both the labelled-line and the hemispheric difference model predict level invariance. This could explain how studies that tested for the role of sound level over a range of high intensities did not see an effect. Moreover, when presented in the free field, sounds also contain interaural level differences and spectral cues, in addition to ITD. For low-frequency sound, listeners rely dominantly on ITD when judging lateral source angle. However, for broadband sound, listeners integrate across all three types of spatial cue (*Wightman and Kistler, 1992*; *Ihlefeld and Shinn-Cunningham, 2011*). Unlike ITDs, interaural level differences and overall sound intensity both decrease with increasing source distance, raising the possibility that for stimuli with high-frequency content, listeners judged softer sounds to be more medial because they interpreted them to be farther away than louder sounds. Further, at low sound intensities, the sound-direction-related notches of the spectral cues at high-frequencies should have been less audible than at higher sound intensities, increasing stimulus ambiguity. A resulting increase in response variability may have obscured the effect of sound intensity on ITD coding. Finally, some historic studies used only two or three listeners, suggesting that they may have been statistically underpowered. Thus, the literature provides insufficient evidence on how ITD-based lateralization varies with sound level near sensation threshold.

## Human perception

Here, we contrasted two competing hypotheses toward the goal of disentangling whether ITD-based human sound localization relies on a labelled-line versus a population rate-place neural code. The labelled-line code hypothesis predicted that the mean perceived direction based on ITD would be intensity invariant, even at intensities close to SL. Using a psychophysical paradigm, we studied lateralization based on ITD as a function of sound intensity in a group of ten normally hearing listeners (experiment 1). Stimuli consisted of low-frequency noise tokens that were bandlimited to cover most of the frequency range where humans can discriminate ITD (*Brughera et al., 2013*; here, corner frequencies from 300 to 1200 Hz, shown in *Figure 2A*). In each one-interval trial, listeners had to indicate perceived laterality across a range of ITDs from −375 to 375 μs. Lateralization was measured as function of SL. To examine how sound intensity affects perceived ITD coding of source direction, we modelled perceived laterality with a nonlinear mixed effect model (NLME) that included fixed effects of ITD and sound intensity as well as a random effect of listener.

*Figure 2B* depicts lateralization performance with spectrally flat noise at two sound intensities for a representative listener (*TCW*). *Figure 2C* shows raw data (circles) and NMLE fits (lines) across all listeners. Error bars show one standard error of the mean across listeners, and shaded ribbons indicate one standard error of the mean fit across listeners. This model predicts 80.6% of the variance in the measured responses and is deemed an appropriate fit of the data. *Table 1* lists all NLME parameters. Perceived laterality scores increased with increasing ITD, as expected. With decreasing sound intensity, percepts were increasingly biased towards midline (compare order of colored lines, magnified in the inset of *Figure 2C*). These trends were supported by the NLME model, which revealed significant effects of ITD (p<0.001; $\alpha_{x1}$) and sound intensity (p<0.001; $\alpha_{y1}$) on the maximal extent of laterality, confirming the predicted trend from the hemispheric difference model and rejecting our null hypothesis. Average pure tone audiometric thresholds affect perceived laterality, albeit mildly (p<0.001; $\alpha_{y2} = 0.01$). Sound intensity did not significantly affect the slope of the psychometric functions (p=0.14; $\alpha_{x2}$).

In a second experiment, we examined whether these results were robust to the spectral details of the stimuli. A caveat of testing spectrally flat noise at low sound intensities is that parts of the spectrum may be inaudible, and this may contribute to the slight but significant effect of audibility on

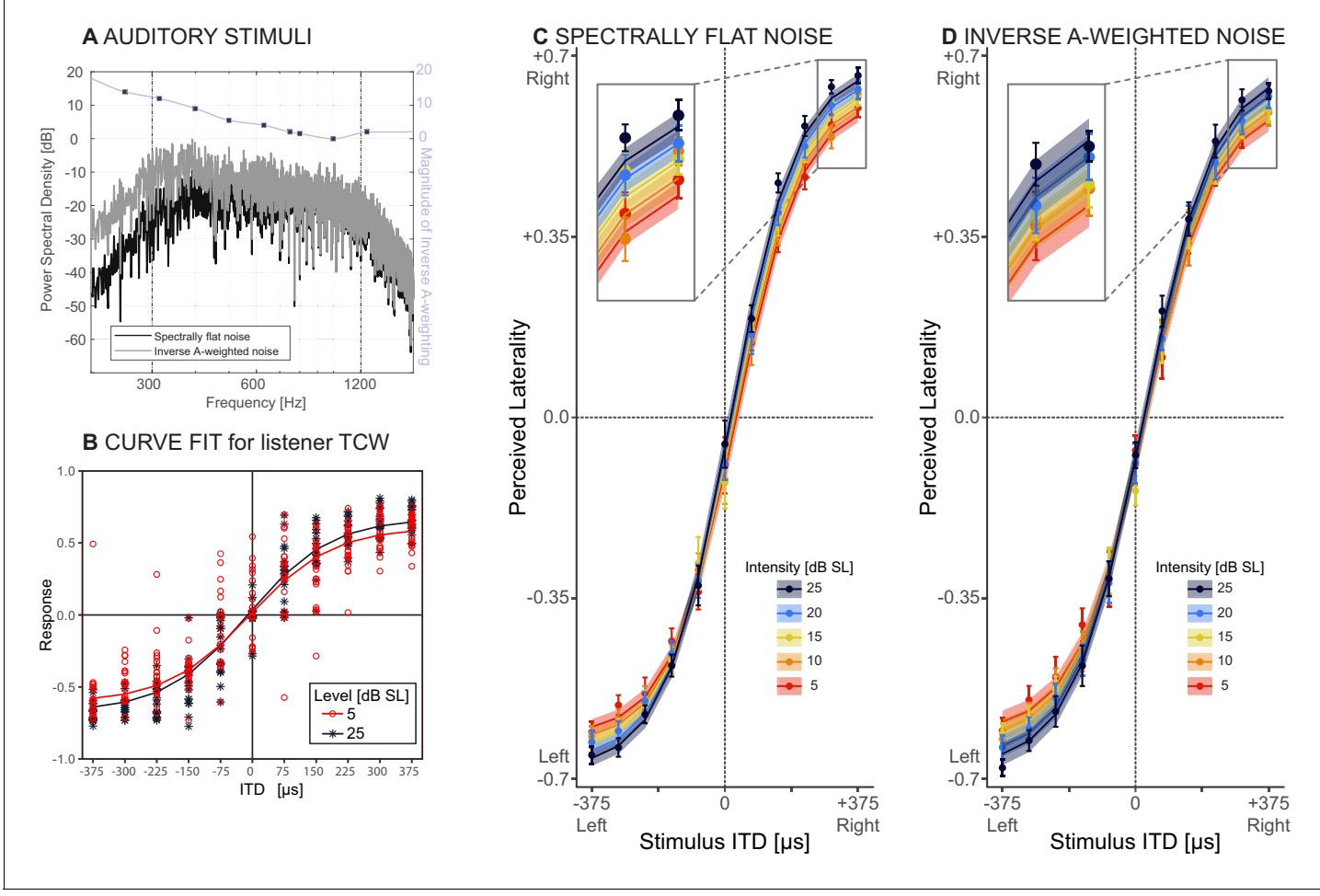

**Figure 2.** Behavioral results. (**A**) Stimuli: spectrally flat noise, used in experiment 1 (dark grey) versus A-weighted noise, tested as a control for audibility in experiment 2 (light grey). The purple line shows the magnitude of the zero-phase inverse A-weighting filter. (**B**) Responses from one representative listener (TCW) across two sound intensities and the corresponding NLME fits for these data. (**C and D**) Perceived laterality as a function of ITD for C) spectrally flat noise (experiment 1) or D) A-weighted noise (experiment 2). Error bars, where large enough to be visible, show one standard error of the mean across listeners. Colors denote sound intensity. Insets illustrate magnified section of the plots. Circles show raw data, lines and ribbons show NLME fits and one standard of the mean.

DOI: https://doi.org/10.7554/eLife.47027.004

laterality ($\alpha_{y2}$). Therefore, the results of experiment 1 could potentially be confounded by the fact that the bandwidth of the audible portion of the noise tokens decreased with decreasing sound intensity. Alternatively, the effect of absolute pure tone detection thresholds that we observe in our normal-hearing listeners may reflect differences in neural function beyond audibility. As a control for perceived stimulus bandwidth, the same listeners were tested again, using inverse A-weighted noises (experiment 2). Inverse A-weighting boosts sound energy at each frequency in rough proportion to the human threshold. Resulting inverse A-weighted sensitivity thus achieves nearly constant sensation level across frequency. All of the original ten listeners from experiment 1 completed experiment 2. Methods were similar as in the first experiment, except that the stimuli consisted of inversely A-weighted noise (compare magnitude spectra in *Figure 2A*). The data and NLME model fits for the second experiment are shown in *Figure 2D* (color key identical to *Figure 2C*), and coefficients are listed in *Table 2*. This second model accounts for 80.4% of the variance in the data, closely fitting the measured responses. All NLME coefficients are significant (p<0.001 for $\alpha_{x1}$; $\alpha_{x2}$; $\alpha_{y1}$ and $\alpha_{y2}$). The intercept coefficient ($\alpha_{x0}$ estimate = −0.60, SE = 0.03, p<0.001) revealed a slight leftward response bias, consistent with a slight narrowband interaural level difference in our stimuli due to precision limits of our test system. The fact that $\alpha_{x2}$ is significant shows that when all noise portions

**Table 1.** Results of Nonlinear Mixed Effects Model for flat-spectrum noise condition.
Note that *Laterality:sound intensity* refers to the NLME weight attributed to acoustic sound intensity of the auditory target. In contrast, Laterality:audibility captures the NLME weight attributed to pure tone audiometric thresholds based on the listeners' perceptual abilities (see Materials and methods for details).

| Description | | Value | Std.error | t-value | p-value | |
|---|---|---|---|---|---|---|
| *Intercept: ITD* | $\alpha_{x0}$ | 0.06 | 0.04 | 1.58 | 0.11 | |
| *Slope: ITD* | $\alpha_{x1}$ | 2.45 | 0.05 | 46.15 | <0.001 | *** |
| *Slope: sound intensity* | $\alpha_{x2}$ | 0.02 | 0.01 | 1.47 | 0.14 | |
| *Laterality: sound intensity* | $\alpha_{y1}$ | 0.05 | 0.01 | 7.59 | <0.001 | *** |
| *Laterality: audibility* | $\alpha_{y2}$ | 0.01 | 0.002 | 4.86 | <0.001 | *** |

10986 degrees of freedom.
DOI: https://doi.org/10.7554/eLife.47027.005

are approximately equally audible, as here, with inverse A-weighted noise, both perceived laterality and the slope linking the change in laterality to ITD decrease with decreasing sound intensity. This is consistent with the interpretation that by controlling for audibility across-frequency, the sensitivity of the task to sound level increases, revealing a medial bias effect not only for the most lateral but also for more medial source angles.

Thus, the results confirm the effect of biasing perceived laterality toward midline with decreasing sound intensity. Therefore, for both spectrally flat noise and A-weighted noise, statistical analyses, which partialed out overall differences between listeners, are inconsistent with a labelled-line model of human sound localization.

## Discussion

Population rate-coding to compute sensory dimension may not be unique to the auditory system. In analogy to sound localization based on the comparison of signals from the two ears (*Figure 3A*), visual depth is computed in the cerebral cortex based on signals from the two eyes (*Figure 3B*; *Poggio, 1995*; *Parker and Cumming, 2001*; *Parker, 2016*). Specifically, in both primary *V1* and extrastriate *V3a* cortex of rhesus macaque monkeys, three types of neurons are thought to encode binocular disparity. 'Tuned-excitatory' neurons respond best to zero spatial disparity between the two eyes, whereas 'near cells' respond more vigorously when an object approaches, increasing crossed disparity between the eyes (*Parker and Cumming, 2001*). Finally, 'far cells' fire more vigorously as uncrossed disparity increases. In *V1*, the most frequently encountered type of binocular neurons are of the tuned-excitatory type. However, in *V3a* the large majority of neurons is stereospecific (*Poggio et al., 1988*) and most neurons are either near or far cells. Functional magnetic resonance imaging experiments on human stereoscopic vision found that unlike *V1* activity, the activity in cortical area *V3a* predicts behavioral performance on tasks involving stereoscopic depth (*Backus et al., 2001*). These observations lead us to propose that in order to compute perceptual space from sensory input, the central nervous system has evolved a canonical computation that is

**Table 2.** Results of Nonlinear Mixed Effects Model for inverse A-weighted noise condition.

| NLME weight | | Value | Std.error | t-value | p-value | |
|---|---|---|---|---|---|---|
| *Intercept: ITD* | $\alpha_{x0}$ | −0.60 | 0.03 | −19.28 | <0.001 | *** |
| *Slope: ITD* | $\alpha_{x1}$ | 2.57 | 0.06 | 46.26 | <0.001 | *** |
| *Slope: sound intensity* | $\alpha_{x2}$ | 0.06 | 0.01 | 4.98 | <0.001 | *** |
| *Laterality: sound intensity* | $\alpha_{y1}$ | 0.04 | 0.01 | 7.10 | <0.001 | *** |
| *Laterality: audibility* | $\alpha_{y2}$ | 0.01 | 0.002 | 3.30 | <0.001 | *** |

10986 degrees of freedom.
DOI: https://doi.org/10.7554/eLife.47027.006

common to different sensory modalities. Specifically, we propose that near and far cells encode visual distance from the fixation plane in a way similar to how *inferior colliculus* neurons encode auditory azimuthal angle away from midline reference: firing rate increases monotonically with distance from perceptual reference anchor or fixation.

We observe that in both the auditory and the visual system, the same cells that are tuned to binaural ITD or binocular disparity also have intensity-response functions. A rate-code based on a population of these cells should cause ambiguities when stimulated below the saturation firing rate, either at low sound intensity or at low contrast (*Figure 3C*). Thus, based on the analogies between the stereo-depth computation and the azimuth-ITD computation, we hypothesized that low visual contrast might affect the computation of depth in a manner analogous to the effect of low sound levels in sound localization—there might be a bias to lower perceived depth at lower contrast (*Figure 3D*). Indeed, one study found such an effect, but only in some observers (*Cisarik and Harwerth, 2008*). A confounding factor in that earlier study is that perceived depth is a complicated neural computation, not only dependent on stereoscopic disparity but also on monocular cues including contrast (*Parker, 2016*). Several studies on depth perception indicate that low contrast is interpreted by the brain as a cue for distance; lower contrast targets are perceived farther away (e.g. *Schor and Howarth, 1986*; *Rohaly and Wilson, 1999*). However, experiments that controlled for low contrast bias demonstrated that low contrast causes perceived depth to shrink, both for near and far deviations from baseline (*Chen et al., 2016*). Thus, there is a link between population rate-coding and stimulus intensity in perceived visual depth as in perceived auditory azimuth, two perceptual spatial dimensions computed by the brain.

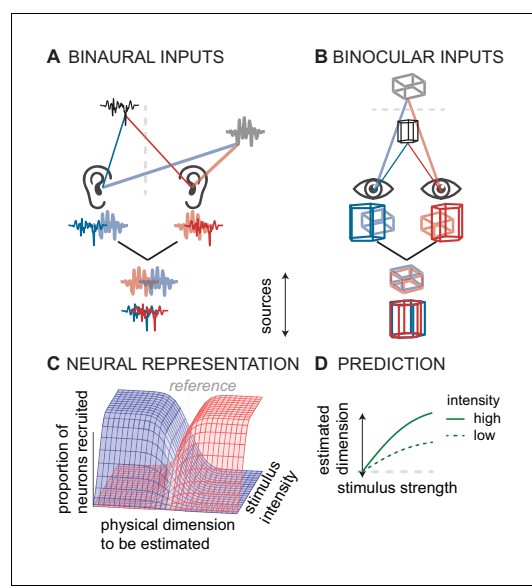

**Figure 3.** Conceptual model of canonical computation of location. (**A**) Computing sound direction requires analysis of the binaural difference between the signals reaching the left and right ear. (**B**) Estimating visual depth hinges on analysis of the binocular disparity between the signals reaching left and right eye. (**C**) For both hearing and vision, the proportion of the neural population that is stimulated (in the *inferior colliculus* or *V3*) depends both on the physical dimension to be estimated (source laterality or source distance) and the intensity of the stimulus (sound intensity or visual contrast). For hearing and vision, ambiguity in this putative neural code predicts D) biased responses at low stimulus intensities (sound intensity or contrast). DOI: https://doi.org/10.7554/eLife.47027.007

To illustrate how rate-based decoding of target location varies with sound intensity, we here chose a rate-coding model that compares firing rates across two populations of neurons, tuned to opposite hemifields. This read-out is a direct realization of the original canonical rate-based model for ITD decoding (*van Bergeijk, 1962*). Alternative rate-code readouts exist (for a recent summary of binaural models, see *Dietz et al., 2018*). Most of these rate-code models rely on subtractive comparisons between populations of neurons that are tuned to opposite hemifields, inherently sharing ambiguous readouts at low suprathreshold sound intensities. In contrast, divisive comparisons between ipsi- and contralaterally tuned neural populations are less likely to predict the observed behavioral bias due to stimulus intensity (*Groh, 2001*). Future work will need to delineate how specific implementations of rate-based readouts shape the intensity-induced bias of sound localization. Moreover, it has been suggested that depending on perceptual task, the mammalian brain could combine place- and rate-codes (*Porter and Groh, 2006*; *Goodman et al., 2013*). For instance, the mammalian auditory pathway may convert place- into rate-codes and vice versa (*Groh et al., 2003*; *Porter and Groh, 2006*). However, downstream from the *inferior colliculus*, rate-coding seems to be maintained, at least in the superior colliculus of rhesus macaque (*Werner-Reiss and Groh, 2008*; *Lee and Groh, 2014*). Moreover, our psychophysical and computational results suggest that for sound localization based on ITD at low sound levels, cortical maps do not play a role.

However, there is good evidence for spatial map-like signals in higher order auditory cortical fields when interaural level differences are present, at medium to high sound levels (*Higgins et al., 2010*). How these interaural-level-difference-based cortical maps influence sound localization behavior is yet to be determined.

An additional factor restricting rate-based readouts is that auditory cortex units display nonlinear rate-intensity functions. For instance, excitatory-excitatory (EE) cells in auditory cortex that are tuned to sound locations near midline are also often tuned for sound intensity (*Semple and Kitzes, 1993*; *Pollak et al., 2002*; *Zhang et al., 2004*; *Razak and Fuzessery, 2010*; *Higgins et al., 2010*). This intensity tuning may complicate rate-based decoding at higher sound intensities. However, it is not apparent at the very low sound intensities needed to explain the perceptual bias observed here. There are additional fascinating findings in the neurophysiological literature regarding frequency and intensity tuning, and interesting correlations between non-monotonicity in the azimuthal and intensity dimensions (*Woods et al., 2006*), but a detailed discussion of these points is beyond the scope of the present behavioral-computational study.

In summary, unlike predictions from a rate-code neuronal readout, labelled-line coding predicts that sound localization is intensity invariant. Our experimental results show that for low frequency noise, where ITDs are the dominant localization cue, and at low sound intensities, sound lateralization based on ITD is not intensity invariant; it becomes increasingly medially biased with decreasing SL. The observed localization bias is overall small in magnitude, showing that the brain can robustly localize based on ITD across a large range of sound intensities. However, this bias is of theoretical importance as it confirms the prediction of a subtractive rate-based neuronal readout. Moreover, our auditory finding parallels a phenomenon of visual fixation bias when calculating visual distance from binocular disparity at low contrast. This casts doubt on the idea that the neural mechanism of ITD-based sound localization and binocular disparity-based visual distance estimation are based on place-based coding. Instead, our perceptual data on auditory localization together with previously published data on visual distance perception are parsimonious with the idea that a population rate-code underlies the brain's computation of location.

## Materials and methods

### Experimental model and subject details

Twelve naïve normal-hearing listeners (ages 18–27, five females) were enrolled in this study and paid for their time. Their audiometric thresholds, as assessed via a calibrated GSI 39 Auto Tymp device (Grason-Stadler), were 25 dB hearing level or better at octave frequencies from 250 to 8000 Hz, and did not differ by more than 10 dB across ears at each octave frequency. This study was approved by and all testing was administered according to the guidelines of the Institutional Review Board of the New Jersey Institute of Technology, protocol F217-14. All listeners gave written informed consent both to participate in the study and to publish the results with confidential listener identity.

### Method details

Listeners were seated in a double-walled sound-attenuating booth (Industrial Acoustics Company) with a noise floor of 20.0 dB SPL (wideband LAFeq). Stimuli were digitally generated in Matlab R2016b (The MathWorks, Inc), D/A converted through an external sound card (Emotiva Stealth DC-1) at a sampling frequency of 192 kHz, with a resolution of 24 bits per sample, and presented to the listener through ER-2 insert earphones (Etymotic Research Inc). The equipment was calibrated using an acoustic mannequin (KEMAR model, G.R.A.S. Sound and Vibration) with a precision of less than ±5 μs ITD and less than ±1 dB interaural level difference. Foam eartips were inserted following guidelines provided by Etymotic Research to encourage equal representation of sounds to both ears and minimize interaural leakage. Each session lasted approximately 60 min. Listeners kept the insert earphones placed inside their ears throughout testing. Insert earphones were replaced by the experimenter after each break. Throughout this study, to generate stimuli, tokens of uniformly distributed white noise were generated and bandpassed using a zero-phase Butterworth filter with 36 dB/octave frequency roll-off, and 3 dB down points at 300 and 1200 Hz. Each noise token was 1 s in duration, including 10 ms long squared cosine ramps at the onset and offset.

## Sensation level measurements

At the beginning of each session, and, as a re-test control, mid-way through each session, each individual listener's SL was measured for the type of sound that was later on used for training and testing, via one run of adaptive tracking. On each one-interval trial of each track, a new noise token was generated and presented diotically. Trials were spaced randomly in time (uniform distribution, inter-token intervals from 3 to 5.5 s). Listeners pressed a button when they heard a sound. No response feedback was given.

On each trial, a response was scored a 'hit' if a listener responded with a button push before the onset of the subsequent trial, and a 'miss' if the listener did not respond during the interval. If a listener's response changed from hit to miss or from miss to hit across sequential trials, this was interpreted as a response reversal. Using one-up-one-down adaptive tracking, the noise intensity was increased or decreased after each reversal, with a step size of 5 dB (decreasing) or 2.5 dB (increasing). Each listener completed ten adaptive-track reversals, with SL threshold equaling the median of the final six reversals. Each SL was used as reference intensity for the subsequent 30 min of testing. If detection thresholds changed between initial test and re-test control by more than 5 dB, this indicated that an insert earphone moved, and the experimenter replaced the earphones. Thresholds generally did not change by more than 5 dB.

## Training

To train listeners on consistently reporting their perception of ITD, using adaptive tracking, listeners matched the perceived laterality of a variable-ITD pointer to that of a fixed-ITD target. Target token intensity was set relative to the listener's own diotic sensation threshold, at 10 or 25 dB SL, and presented with 0 dB interaural level difference. The pointer intensity was fixed at 25 dB SL. Target ITDs spanned the range from −375 to 375 µs, in 75 µs steps. Target ITDs and SLs were randomly interleaved across runs, but held fixed throughout each adaptive run. In each two-interval trial of a run, the pointer token was presented in the first, and the target token in the second interval. The start ITD of the pointer token at the beginning of each run equaled 0 µs. Using a hand-held controller (Xbox 360 wireless controller for Windows, Microsoft Corp.), listeners adjusted the ITD of the pointer token. Specifically, listeners pushed the directional keys (D-pad) either to the left or right in order to move and match the pointer direction with that of the target sound. When a listener indicated a left- or right-ward response, the pointer ITD was decreased or increased. Initial ITD step size equaled 100 µs, then 50 ±5 µs (uniformly distributed) after the first reversal. By the end of the second reversal, ITD step size was reduced to 25 ±5 µs (uniformly distributed) and remained the same for all of the following reversals. Listeners were instructed to 'home in' on the target by moving the pointer initially to a position more lateral than the target, then more medial than the target with the goal of centering on the target. No response feedback was provided. A run was completed after a listener had completed a total of five adaptive-tracking reversals. For each target ITD, the matched pointer ITD was estimated by averaging the pointer ITDs of the final two reversals. Each listener performed three sessions of training: In the first session only a subset of target ITDs were presented (−375,−150, 0, 150 and 375 µs), whereas the two following sessions included all of the eleven ITDs. Per training session, each ITD was presented once at 10 and 25 dB SL, for a total of 54 adaptive tracking runs across all training sessions. To familiarize listeners with the experimental task (described below), at the end of second and third sessions of training listeners performed an additional 5 blocks of the experimental testing task, without response feedback. These task training data were not used for statistical analysis.

To assess whether listeners could reliably report their lateralization percepts, training performance was evaluated for each listener by calculating the Pearson correlation coefficient between target ITD and matched pointer ITD in the final training session. Criterion correlation equaled 0.9 (N = 11 ITDs, significance level = 0.01, power = 0.95). Ten listeners reached criterion, suggesting that they were able to consistently report where they perceived the sounds based on ITD. Two of the originally recruited twelve listeners failed to reach training criterion ($R^2$ = <0.84, 0.87>) and were excluded from testing.

## Testing

Using the method of fixed stimuli, we tested lateralization in two experiments. Except for the stimuli, which consisted of spectrally flat noise tokens in experiment 1 and A-weighted noise tokens in experiment 2, the methods were similar across the two experiments. Noise tokens were generated from a statistically similar noise distribution as those presented during both SL measurements and training (see Overall Design). A touchscreen monitor (Dell P2314T) displayed the response interface at about 40 cm distance from the listener. Using a precise touch stylus (MEKO Active Fine Point Stylus 1.5 mm Tip), listeners indicated perceived laterality of noise in a one-interval task. Noise tokens were presented at 5, 10, 15, 20, and 25 dB SL. ITDs varied randomly from trial to trial, in 75 µs steps spanning the range from −375 µs to 375 µs. On each trial, a new token of noise was generated. Each listener performed 20 blocks of 55 trials each (11 ITDs at each of the five sound intensities), with SL measured both before the first and the eleventh block. ITDs and sound intensity were randomly interleaved from trial to trial such that each combination of ITD and sound intensity was presented once before all of them were repeated in a different random order.

## Models

We estimate the combined effects of ITD and sound intensity on predicted source laterality both in avian labelled-line type units and in binaurally sensitive units of a mammalian auditory system. The sound intensities where we expect to see an effect of overall sound level fall below 30 dB SPL, because only in this range would most auditory neurons fire below saturation, allowing us to disambiguate labelled-line versus hemispheric rate-difference coding. However, scant data exist for either type of unit at sound pressure levels below 30 dB SPL. We identified two prior studies that have measured neural discharge rate as a function of ITD at these very low sound intensities. Both studies used noise as acoustic stimuli, and the neural response statistics they report are thus suitable for estimating what type of information would be available to either type of coding mechanism with the type of noise stimuli that human listeners lateralized in the behavioral experiments here.

One study in barn owl shows that the output functions of *nucleus laminaris* neurons can be modeled through interaural cross-correlation functions, even at very low sound intensities (*Peña et al., 1996*). That study reports Pearson correlation coefficients between the neural response function of *nucleus laminaris* units at 50 dB SPL versus all other tested sound levels. To reconstruct the spatial information realistically available from the output of labelled-line neurons, across both a range of −375 to 375 µs ITD in 20 µs steps, we first constructed biologically plausible interaural cross-correlation functions at 50 dB SPL and then added internal noise to the resulting curves to mimic the Pearson correlation coefficients reported by *Peña et al. (1996)*. Our model predictions pertain to sound intensities spanning the range from 10 to 70 dB SPL, similar to previous work (*Peña et al., 1996*). Due to overall scarcity of available data at low dB SPL, here we use firing rate characteristics for unit # 0123795–530.02 (*Peña et al., 1996*) with a nominal best frequency of 1 kHz. To generate the acoustic inputs to the labelled-line model, we initially generated a Gaussian noise token, duplicated it and introduced a variable ITD, spanning a range from −375 to 375 µs, with 20 µs step size and 0 dB interaural level difference. To simulate ITD information available after cochlear processing, we then processed both noises with a 1/3-octave wide bandpass filter with 24 dB/octave frequency rolloff, followed by half-wave rectification and low-pass filtering at 1500 Hz. We then simulated internal noise by adding uniformly distributed dichotic noise tokens with mean spontaneous firing rates of 5% of the root mean square value of the signal, resulting in left (L) and right (R) inputs to the binaural cross-correlation neurons, called $x_L(t)$ and $x_R(t)$. To establish 50 dB SPL reference functions, at each simulated ITD, we then calculated the binaural cross-correlation function $cc(\tau)$ of $x_L(t)$ and $x_R(t)$, as follows: $cc(\tau) = 300 + (450 - 300)\frac{\int_{-\infty}^{+\infty} x_L(t)x_R(t+\tau)dt}{max|\int_{-\infty}^{+\infty} x_L(t)x_R(t+\tau)dt|}$, with $\tau$ signifying the best ITD of each neuron, and extrema scaled such that $cc(\tau)$ spans a range from 300 to 450 spikes/sec, approximating *nucleus laminaris* firing rates at 50 dB SPL (*Peña et al., 1996*). To simulate non-sound driven neural discharge, we then added uniformly distributed random noise $\hat{cc}_{ref}(\tau) = cc(\tau) + U(0, \mu)$, with a mean discharge of $\mu = 5$ spikes/sec, (*Peña et al., 1996*). The resulting signal is our reference cross-correlation function at 50 dB SPL, called $cc_{ref}(\tau)$, shown in *Figure 1A* as yellow bold line for a representative simulated neuron.

For each sound level and ITD, we then statistically reconstructed a family of interaural cross-correlation functions that match the originally reported functions (*Peña et al., 1996*). Specifically, we added scaled dichotic uniformly distributed noise tokens $n_L(t) \leftarrow U(0, \mu)$ and $n_R(t) \leftarrow U(0, \mu)$ to the $x_L(t)$ and $x_R(t)$, such that the monaural inputs to the binaural cross-correlation functions equal $\hat{x}_{L,R}(t) = \alpha x_{L,R}(t) + \sqrt{1 - \alpha^2} n_{L,R}(t)$. The resulting cross-correlation function for each sound level and ITD is then $\hat{cc}(\tau) = (x_L \star x_R)(\tau)$, shown for a representative neuron in *Figure 1A* as blue, brown and red lines corresponding to 70, 30 and 10 dB SPL. We then searched through the space of scaling coefficients $\alpha$ until the Pearson correlation coefficient between $cc_{ref}(\tau)$ and $\hat{cc}(\tau)$ matched the coefficients originally reported by *Peña et al. (1996)* with a precision error of less than 10%.

To estimate predicted sound laterality as a function of sound intensity for these simulated labelled-line neurons, at each intensity, we then identified the $\tau$ where $\hat{cc}(\tau) = max(\hat{cc}(\tau))$. For each sound level and ITD, we calculated predicted sound laterality in 100 repetitions of these simulations. *Figure 1C* shows mean estimated laterality across these 100 simulations, with ribbons showing one standard error of the mean across simulations.

To estimate source laterality based on rate-coding, we assayed the mammalian auditory system, where one previous study reports firing statistics for 81 *inferior colliculus* units in rhesus macaque as a function of ITD and over a wide range of sound intensities, including very low sound intensities (*Zwiers et al., 2004*). From the previously published linear regression parameters, we initially reconstruct linear regression functions linking ITD, sound intensity and firing rate (*Zwiers et al., 2004*). However, while linear regression fits afford statistical convenience, they cannot fully capture the sigmoidally shaped firing rate functions in mammalian *inferior colliculus* units. Therefore, we multiplied the original linear reconstructions with sigmoid functions. Specifically, consistent with prior literature, each simulated sigmoidal output function saturates over a 30 dB dynamic range, has linear growth over the physiologically plausible range of contralateral ITDs, has a threshold between uniformly distributed between 0 and 10 dB SPL, and a spontaneous non-sound-evoked discharge of between 2 and 10 spikes/second (e.g. *Ramachandran et al., 1999*).

The inset of *Figure 1B* shows a representative simulated *inferior colliculus* unit (color denotes sound intensity, dark shading shows contralateral responses), whereas *Figure 1B* shows the differences in firing rates for contra minus ipsi-lateral simulated firing rates, averaged across all 81 simulated *inferior colliculus* units. From these resulting differences in contra versus ipsi firing rates we calculated, collapsed across sound intensities from 0 to 80 dB SPL, the probability density of the firing rate for each *inferior colliculus* unit as a function of source ITD. Assuming an ideal observer, we then classified the sound azimuth as a function of sound intensity via maximum likelihood estimation. To calculate the mean and variance of predicted ITD as a function of sound intensity, we then ran a bootstrapping analysis, sampling with replacement 100 times. *Figure 1D* shows the across-simulation average predicted source laterality, with ribbons showing one standard error of the mean across simulations.

## Quantification and statistical analysis

Growth curve analysis was used to analyze perceived laterality scores as a function of ITD and sound intensity. For each of the two noise conditions, the perceived laterality scores were fitted with an NLME model. The model included fixed effects $\alpha$ and random effects $\beta$. *Equation 1* describes a sigmoidal function linking ITD to perceived laterality, with a score from left (−1) to right (1). The effect of sound intensity on the maximal extent of lateralization is $\alpha_{y1}$. To factor out across-listener differences in absolute hearing thresholds, for each listener, we calculated the pure tone average (PTA) detection threshold in quiet, averaged across ears, and across 500 and 1000 Hz. Weight $\alpha_{y2}$ models the contribution of PTA. The slope terms are $\alpha_{x1}$ for perceived laterality changes attributed to ITD, and $\alpha_{x2}$ for laterality-ITD slopes attributed to sound intensity. Our stimuli were initially calibrated to have a broadband interaural level difference of 0 dB. However, because the transfer function of our sound card was not perfectly flat across frequency, fluctuations of ±1 dB interaural level difference occurred across frequency, on the same order of magnitude as the minimal threshold for human interaural level difference discrimination (*Francart and Wouters, 2007*). Thus, parameter $\alpha_{x0}$ factors out central response bias from the lateralization scores. Random effects of individual differences across listeners were used to model both the maximal extent of lateralization, $\beta_{y0,listener}$, and the perceived midline, $\beta_{x0,listener}$, centering the sigmoid (*Equation 1*):

$$response \sim \frac{\alpha_{y2} \times PTA + \alpha_{y1} \times intensity + \beta_{y0,listener}}{1 + e^{-\left[\alpha_{x2} \times intensity + \alpha_{x1} \times \left(ITD - \alpha_{x0} - \beta_{x0,listener}\right)\right]}} - 0.5 \qquad (1)$$

To better conform with the assumptions of the NLME model, prior to fitting, ITD and sound intensity parameters were scaled by subtracting the mean stimulus value, and dividing by the standard deviation of stimulus parameters, resulting in distributions of stimulus parameters with zero-mean and a variance of one. Laterality scores were then fitted using these normalized parameters, with the nlme package, programmed in RStudio 1.1 for Windows (RStudio Inc, Boston, MA, USA).

## Data and software availability

All data and analysis code are available at Dryad (http://doi.org/10.5061/dryad.t8c381f) .

# Additional information

## Funding

| Funder | Grant reference number | Author |
|---|---|---|
| New Jersey Health Foundation | PC 24-18 | Antje Ihlefeld |

The funders had no role in study design, data collection and interpretation, or the decision to submit the work for publication.

## Author contributions

Antje Ihlefeld, Conceptualization, Software, Formal analysis, Supervision, Funding acquisition, Methodology, Writing—original draft, Project administration, Writing—review and editing; Nima Alamatsaz, Software, Formal analysis, Investigation, Writing—review and editing; Robert M Shapley, Conceptualization, Methodology, Writing—review and editing

## Author ORCIDs

Antje Ihlefeld (iD) https://orcid.org/0000-0001-7185-5848
Nima Alamatsaz (iD) https://orcid.org/0000-0003-3374-3663

## Ethics

Human subjects: This study was approved by and all testing was administered according to the guidelines of the Institutional Review Board of the New Jersey Institute of Technology, protocol F217-14. All listeners gave written informed consent both to participate in the study and to publish the results with confidential listener identity.

## Decision letter and Author response

Decision letter https://doi.org/10.7554/eLife.47027.012
Author response https://doi.org/10.7554/eLife.47027.013

# Additional files

## Supplementary files

• Transparent reporting form DOI: https://doi.org/10.7554/eLife.47027.008

## Data availability

The computational models, coded in Matlab, and all raw data generated or analysed during this study together with the statistical model, coded in R, have been provided via Dryad (http://doi.org/10.5061/dryad.t8c381f).

The following dataset was generated:

| Author(s) | Year | Dataset title | Dataset URL | Database and Identifier |
|---|---|---|---|---|
| Ihlefeld A, Alamatsaz N, Shapley RM | 2019 | Data from: Human sound localization depends on sound intensity: implications for sensory coding | http://doi.org/10.5061/dryad.t8c381f | Dryad Digital Repository, 10.5061/dryad.t8c381f |

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
