## [Decision Letter]

Thank you for submitting your article "Population rate coding predicts correctly that human sound localization depends on sound intensity" for consideration by *eLife*. Your article has been reviewed by three peer reviewers, including Catherine Emily Carr as the Reviewing Editor and Reviewer #1, and the evaluation has been overseen by Barbara Shinn-Cunningham as the Senior Editor. The following individuals involved in review of your submission have agreed to reveal their identity: Heather Read (Reviewer #2) and Jennifer M Groh (Reviewer #3).

The reviewers have discussed the reviews with one another and the Reviewing Editor has drafted this decision to help you prepare a revised submission.

Summary:

This is a combined modeling and behavioral study of human sound localization that supports rate-coding models. Since sound levels affect the firing rate of individual neurons, sometimes changes in intensity can be confused with changes in location. The experiments investigate psychophysical evidence for this predicted effect, with positive results.

The authors use a hemispheric difference rate coding model of a range of interaural time delays (ITD) to predict that sound localization would be biased towards midline locations at low sound levels. Though the hemispheric rate coding model is simplistic and has been explored before, it is a useful way to raise the question of whether such biases might exist perceptually in auditory systems as they do in visual systems.

Suggestions for improvement mostly concern additions and clarifications, not new experiments. In general questions from the review should be answered by changes to the text. We number major suggestions below.

1) Clarification of response types:

Towards the end, the Discussion section explains appropriately how "both the auditory and the visual system, the same cells that are tuned to binaural ITD or binocular disparity also have intensity-response functions….." that should cause ambiguities (biases) in localizing sensory objects near the sensory midline. Figure 3 illustrates this principle with monotonic intensity dependence of neural response rate functions for ITD in central inferior colliculus. Thus, this part of the discussion is appropriately focused on some of the universal principles for neurometric and psychometric localization of sensory objects/stimuli in both visual and auditory systems, and their intensity/contrast dependence.

However, in the first paragraph of the Discussion section, some details are given regarding how visual cortical neurons fall into three types: (1) "Tuned-excitatory" neurons [that] respond best to zero spatial disparity; (2) "near cells" [that] respond more vigorously when [ocular] disparity[is crossed] (Parker and Cumming. (2001)); (3) and "far cells" when ocular disparity is uncrossed. This paragraph goes on to say that secondary visual cortices (V3) respond by and large to binocular cues (as opposed to monocular) cues. Do those binocular responses display biases towards the sensory field midline at low contrast? By analogy IT cortical neurons are biased towards central 0-6 degrees visual angle, is the IT central biased and tuning binaural and contrast dependent?

It would be useful to explain that neural responses in the auditory cortices are also distributed into regions and cortical fields according to binaural types. Auditory cortices have contralateral tuned-excitatory neurons that respond best to contralateral location disparity (e.g. level difference or ITD difference) and auditory cortices also have "tuned-excitatory" neurons that respond best to zero spatial disparity. The latter have been called (EE/F) or "primarily binaural" or "two-way intensity" tuned (Semple and Kitzes, 1993b; Pollak et al., 2002; Zhang et al., 2004; Razak and Fuzessery, 2010; Higgins et al., 2010). Notably, many if not all of the latter studies find that the "tuned-excitatory" neurons observed in primary (A1) or secondary ventral auditory cortices respond optimally at lower sound intensities. Hence, this raises an interesting twist namely, that neurons also can encode sound location/azimuth position near the midline in spite of the "ambiguities" at low sound intensities possibly by using spatial cues other than ITD.

2) There is some confusion about what the results actually are.a) What is the difference between "audibility" and "intensity" in the statistical assessment (subsection “Human perception”, Table 1 and Table 2) What does audibility mean in this context? If it co-varies with intensity, why is it included as a term in the model? Is it a means of incorporating the audiograms of individual subjects?

b) Aren't changes to the slope of the psychometric function as a function of intensity the key result? Yet, that was significant for the second experiment but not the first (although it was close). I think it would be reasonable to say the first experiment didn't quite reject the null hypothesis but showed a trend, so a second experiment was done that controlled an additional aspect of the stimuli in the task, and the same subjects repeated the study, at which point strongly significant results were then obtained. The authors should be commended for including both experiments in the results.

c) Re: experiment 2: the description in subsection “Human perception” is clear in explaining that in experiment 2, the A-weighting equalizes the perceived bandwidth of the sounds across intensities, but is followed by: "As a control for stimulus audibility". If I have understood the lead-in correctly, this sentence would make more sense to this reader as: "As a control for perceived bandwidth".

3) Computational context:

It would be useful to explain that although this study supports the intensity dependence of sound localization and potentially hemispheric models for the corresponding neural code alternative scenarios may exist. For example, is it not possible that the mammalian brain has both place-codes and rate codes in different parts that are called upon under different circumstances?

The computational underpinnings behind this experiment are perhaps more subtle than currently described. A previous study by Porter and Groh, 2006) lays out a few possible models for converting between place and rate codes and explicitly considers the dependence of neural firing patterns on both sound intensity and any other non-spatial factors (see Porter and Groh Figure 4 and page 319 for description of the models and the evidence for and against).

Factors to consider are: is the proposed readout a subtractive or divisive comparison between left and right channels? How would the predicted effect of sound intensity differ depending on that computational combination rule? Might the relatively small effect of intensity in this task reveal that the brain is doing a pretty good job of handling this under a range of conditions, but it is not perfect?

Other factors to consider are that the place codes may have effects of intensity as well, depending on how they are read out. For example, a weighted-sum readout of a place code will be sensitive to intensity signals. A weighted-average readout, however, incorporates a normalization mechanism that removes the intensity signal. A winner-take-all readout is also robust to intensity. I don't think the authors need to go into extensive detail, but a few qualifying phrases at well-placed locations in the manuscript to alert the reader that the comparison is not a matter of place codes vs rate codes per se but place-codes-and-their-readouts vs. rate-codes-and-their-readouts would go a long way.

References to include:

The Porter and Groh, (2006) study mentioned above is relevant for inclusion. In addition, Salminen and colleagues have conducted a number of studies arguing that the human brain does indeed use a rate code for sound location. In the primate brain, studies by Werner-Reiss and Groh, (2008, auditory cortex) and Groh et al., (2003, inferior colliculus) are relevant. I also recommend including citations to the literature on non-monotonic rate-level functions in the auditory pathway: this term refers to neurons whose responses go up and then *down* as a function of increasing sound intensity. The existence of these neurons may be part of the way in which the brain produces largely accurate sound localization *despite* mixed sensitivity to intensity and laterality in individual neurons.

[Editors' note: further revisions were requested prior to acceptance, as described below.]

Thank you for resubmitting your work entitled "Population rate-coding predicts correctly that human sound localization depends on sound intensity" for further consideration at *eLife*. Your revised article has been favorably evaluated by Barbara Shinn-Cunningham (Senior Editor), a Reviewing Editor, and two reviewers.

The manuscript has been improved but there are some remaining issues that need to be addressed before acceptance, as outlined below:

After consultation among the reviewers regarding multiple details of the underlying neurophysiology as covered in the Discussion section, the consensus recommendation is that the authors add something along the lines that while their study was motivated by the notion that there's no auditory map, it is entirely possible that there are sufficient map-like signals in higher order auditory cortical fields to support perception. Citations for this latter point include: Higgins et al., 2010, and Woods et al., 2006.

There are additional fascinating findings in the neurophysiological literature regarding frequency and intensity tuning, and interesting correlations between non-monotonicity in the azimuthal and intensity dimensions, but a detailed discussion of these points is probably beyond the scope for the present behavioral/computational study.

---

## [Author Response]

Summary:This is a combined modeling and behavioral study of human sound localization that supports rate-coding models. Since sound levels affect the firing rate of individual neurons, sometimes changes in intensity can be confused with changes in location. The experiments investigate psychophysical evidence for this predicted effect, with positive results.The authors use a hemispheric difference rate coding model of a range of interaural time delays (ITD) to predict that sound localization would be biased towards midline locations at low sound levels. Though the hemispheric rate coding model is simplistic and has been explored before, it is a useful way to raise the question of whether such biases might exist perceptually in auditory systems as they do in visual systems.Suggestions for improvement mostly concern additions and clarifications, not new experiments. In general questions from the review should be answered by changes to the text. We number major suggestions below.

Thank you for your thorough and constructive feedback. We now explicitly discuss rate-based readouts from the hemispheric difference model in the context of other rate-based models. We appreciate the pointers to prior decoding models of sound location by Groh and colleagues and have added those to the discussion. We now connect to the spatial adaptation work by Salminen and colleagues, which we previously missed.

1) Clarification of response types:towards the end, the Discussion section explains appropriately how "both the auditory and the visual system, the same cells that are tuned to binaural ITD or binocular disparity also have intensity-response functions….." that should cause ambiguities (biases) in localizing sensory objects near the sensory midline. Figure 3 illustrates this principle with monotonic intensity dependence of neural response rate functions for ITD in central inferior colliculus. Thus, this part of the discussion is appropriately focused on some of the universal principles for neurometric and psychometric localization of sensory objects/stimuli in both visual and auditory systems, and their intensity/contrast dependence.

Thank you.

However, in the first paragraph of the Discussion section, some details are given regarding how visual cortical neurons fall into three types: (1) "Tuned-excitatory" neurons [that] respond best to zero spatial disparity; (2) "near cells" [that] respond more vigorously when [ocular] disparity[is crossed] (Parker and Cumming. (2001)); (3) and "far cells" when ocular disparity is uncrossed. This paragraph goes on to say that secondary visual cortices (V3) respond by and large to binocular cues (as opposed to monocular) cues. Do those binocular responses display biases towards the sensory field midline at low contrast? By analogy IT cortical neurons are biased towards central 0-6 degrees visual angle, is the IT central biased and tuning binaural and contrast dependent?

If somebody has looked at these intriguing questions, we are not aware of the work. We have searched the literature for contrast effects of this sort and could not find data that speaks to the questions.

It would be useful to explain that neural responses in the auditory cortices are also distributed into regions and cortical fields according to binaural types. Auditory cortices have contralateral tuned-excitatory neurons that respond best to contralateral location disparity (e.g. level difference or ITD difference) and auditory cortices also have "tuned-excitatory" neurons that respond best to zero spatial disparity. The latter have been called (EE/F) or "primarily binaural" or "two-way intensity" tuned (Semple and Kitzes, 1993b; Pollak et al., 2002; Zhang et al., 2004; Razak and Fuzessery, 2010; Higgins et al., 2010). Notably, many if not all of the latter studies find that the "tuned-excitatory" neurons observed in primary (A1) or secondary ventral auditory cortices respond optimally at lower sound intensities. Hence, this raises an interesting twist namely, that neurons also can encode sound location/azimuth position near the midline in spite of the "ambiguities" at low sound intensities possibly by using spatial cues other than ITD.

Thank you for raising this interesting twist. We now discuss this point explicitly (Discussion section). We agree with the reviewers that EE cells in auditory cortex that are tuned to the midline are also often tuned for sound intensity. However, this intensity tuning is not apparent at the very low sound intensities needed to explain the bias observed here. In fact the first paper by Semple and Kitzes, (1993) shows clearly a max response at 40 dB SPL and significant responses in the EE cells up to 80 dB SPL, a pattern supported by the later papers (Pollak et al., 2002; Zhang et al., 2004; Razak and Fuzessery, 2010; Higgins et al., 2010). That is, the tuned-excitatory neurons that encode the midline respond best not at the low sound levels where we observed a midline bias, but at sound levels 10X or 100X louder. These neurons would not be expected to be engaged in our task.

2) There is some confusion about what the results actually are.a) What is the difference between "audibility" and "intensity" in the statistical assessment (subsection “Human perception”, Table 1 and Table 2) What does audibility mean in this context? If it co-varies with intensity, why is it included as a term in the model? Is it a means of incorporating the audiograms of individual subjects?

The latter, it is a way of incorporating the audiograms of individual subjects into the statistical model. To keep readers from wondering whether overall audiograms could have caused the observed bias at our very soft sound intensities, we include the audiogram thresholds in the NLME fits. Reassuringly, the NLME parameters linking ITD and sound intensity to perceived laterality do not change appreciably when we include the subject’s pure tone audiometric thresholds. To make this point clear, we have reworded the caption of Table I.

b) Aren't changes to the slope of the psychometric function as a function of intensity the key result? Yet, that was significant for the second experiment but not the first (although it was close). I think it would be reasonable to say the first experiment didn't quite reject the null hypothesis but showed a trend, so a second experiment was done that controlled an additional aspect of the stimuli in the task, and the same subjects repeated the study, at which point strongly significant results were then obtained. The authors should be commended for including both experiments in the results.

Based on our modeling results, we expected to see bigger perceptual changes at lateral as compared to medial angles, and this is indeed what we saw in both experiments (maximal extent of lateralization varied with sound intensity). In addition, experiment 2 revealed a change in slope, meaning by controlling for audibility across-frequency, the effect is also visible at more medial source angles. We now point this out in the Results section.

c) Re: experiment 2: the description in subsection “Human perception” is clear in explaining that in experiment 2, the A-weighting equalizes the perceived bandwidth of the sounds across intensities, but is followed by: "As a control for stimulus audibility". If I have understood the lead-in correctly, this sentence would make more sense to this reader as: "As a control for perceived bandwidth".

Thank you for this suggestion. Reworded.

3) Computational context:It would be useful to explain that although this study supports the intensity dependence of sound localization and potentially hemispheric models for the corresponding neural code alternative scenarios may exist. For example, is it not possible that the mammalian brain has both place-codes and rate codes in different parts that are called upon under different circumstances?

We now talk about this possibility in the Discussion section.

The computational underpinnings behind this experiment are perhaps more subtle than currently described. A previous study by Porter and Groh, 2006) lays out a few possible models for converting between place and rate codes and explicitly considers the dependence of neural firing patterns on both sound intensity and any other non-spatial factors (see Porter and Groh Figure 4 and page 319 for description of the models and the evidence for and against).

Thank you for pointing out how these prior studies connect with the current work. We have now expanded the discussion of rate- vs place-based readouts. We mention the possibility that rate-based could be converted into place-based readouts (Discussion section).

Factors to consider are: is the proposed readout a subtractive or divisive comparison between left and right channels? How would the predicted effect of sound intensity differ depending on that computational combination rule? Might the relatively small effect of intensity in this task reveal that the brain is doing a pretty good job of handling this under a range of conditions, but it is not perfect?

We now explain the predicted effects of sound intensity based on subtractive vs divisive readout (Discussion section). We agree with you that the small effect of intensity suggests that sound localization is largely robust across a range of conditions and now comment on this explicitly in the Discussion section.

Other factors to consider are that the place codes may have effects of intensity as well, depending on how they are read out. For example, a weighted-sum readout of a place code will be sensitive to intensity signals. A weighted-average readout, however, incorporates a normalization mechanism that removes the intensity signal. A winner-take-all readout is also robust to intensity. I don't think the authors need to go into extensive detail, but a few qualifying phrases at well-placed locations in the manuscript to alert the reader that the comparison is not a matter of place codes vs rate codes per se but place-codes-and-their-readouts vs. rate-codes-and-their-readouts would go a long way.References to include:The Porter and Groh, (2006) study mentioned above is relevant for inclusion. In addition, Salminen and colleagues have conducted a number of studies arguing that the human brain does indeed use a rate code for sound location. In the primate brain, studies by Werner-Reiss and Groh, (2008, auditory cortex) and Groh et al., (2003, inferior colliculus) are relevant.

Included.

I also recommend including citations to the literature on non-monotonic rate-level functions in the auditory pathway: this term refers to neurons whose responses go up and then down as a function of increasing sound intensity. The existence of these neurons may be part of the way in which the brain produces largely accurate sound localization despite mixed sensitivity to intensity and laterality in individual neurons.

We now mention that non-monotonic rate-level functions would complicate the read-out (Discussion section). We also point out the fact that these non-monotonicities mostly appear to arise at sound levels that are outside the range of sound intensities for which the hemispheric difference model predicts a behavioral bias.

[Editors' note: further revisions were requested prior to acceptance, as described below.]

The manuscript has been improved but there are some remaining issues that need to be addressed before acceptance, as outlined below:After consultation among the reviewers regarding multiple details of the underlying neurophysiology as covered in the Discussion section, the consensus recommendation is that the authors add something along the lines that while their study was motivated by the notion that there's no auditory map, it is entirely possible that there are sufficient map-like signals in higher order auditory cortical fields to support perception. Citations for this latter point include: Higgins et al., 2010, and Woods et al., 2006.There are additional fascinating findings in the neurophysiological literature regarding frequency and intensity tuning, and interesting correlations between non-monotonicity in the azimuthal and intensity dimensions, but a detailed discussion of these points is probably beyond the scope for the present behavioral/computational study.

We greatly appreciate your thoughtful comments and have added two statements to this effect. In the Discussion section we have added the following sentences:

“Moreover, our psychophysical and computational results suggest that for sound localization based on ITD at low sound levels, cortical maps do not play a role. However, there is good evidence for spatial map-like signals in higher order auditory cortical fields when interaural level differences are present, at medium to high sound levels (Higgins et al., 2010). How these ILD-based cortical maps influence sound localization behavior is yet to be determined.”

In the Discussion section we now also state:

“There are additional fascinating findings in the neurophysiological literature regarding frequency and intensity tuning, and interesting correlations between non-monotonicity in the azimuthal and intensity dimensions (Woods et al., 2006), but a detailed discussion of these points is beyond the scope of the present behavioral-computational study.”